# Filamin A Regulates Cardiovascular Remodeling

**DOI:** 10.3390/ijms22126555

**Published:** 2021-06-18

**Authors:** Sashidar Bandaru, Chandu Ala, Alex-Xianghua Zhou, Levent M. Akyürek

**Affiliations:** 1Division of Clinical Pathology, Sahlgrenska Academy Hospital, 413 45 Gothenburg, Sweden; sashidar.bandaru@medkem.gu.se; 2Department of Laboratory Medicine, Institute of Biomedicine, Sahlgrenska Academy, University of Gothenburg, 405 30 Gothenburg, Sweden; alachandu225@gmail.com (C.A.); Alex.Zhou1@astrazeneca.com (A.-X.Z.)

**Keywords:** actin-binding, cell signaling, cytoskeleton, transcription

## Abstract

Filamin A (FLNA) is a large actin-binding cytoskeletal protein that is important for cell motility by stabilizing actin networks and integrating them with cell membranes. Interestingly, a *C*-terminal fragment of FLNA can be cleaved off by calpain to stimulate adaptive angiogenesis by transporting multiple transcription factors into the nucleus. Recently, increasing evidence suggests that FLNA participates in the pathogenesis of cardiovascular and respiratory diseases, in which the interaction of FLNA with transcription factors and/or cell signaling molecules dictate the function of vascular cells. Localized *FLNA* mutations associate with cardiovascular malformations in humans. A lack of FLNA in experimental animal models disrupts cell migration during embryogenesis and causes anomalies, including heart and vessels, similar to human malformations. More recently, it was shown that FLNA mediates the progression of myocardial infarction and atherosclerosis. Thus, these latest findings identify FLNA as an important novel mediator of cardiovascular development and remodeling, and thus a potential target for therapy. In this update, we summarized the literature on filamin biology with regard to cardiovascular cell function.

## 1. Introduction

Actin-binding cytoskeletal proteins are involved in the formation and maintenance of cell shape and morphology in response to external stimuli from surrounding connective tissue [1]. Filamins are one of the actin-binding proteins that mediate dynamic remodeling during cell movement. Recently, filamins have been shown to be involved in cell signaling and transcription. The filamin family has three members, i.e., filamin A (FLNA), filamin B (FLNB), and filamin C (FLNC). These isoforms exhibit 70% homology with their amino acids and 45% homology in hinge 1 (H1) and hinge 2 (H2) domains [2]. FLNA is the most studied isoform of filamins. Both the human *FLNA* and mouse *FLNA* gene are located on chromosome X. Human *FLNB* and *FLNC* genes are located on chromosomes 3 and 7, respectively, whereas mouse *FLNB* and *FLNC* genes are located on chromosomes 14 and 6, respectively [3]. Although FLNA and FLNB are very similar to one another, their human mutations at different genomic positions result in a wide variety of clinical phenotypes. Filamin isoforms are expressed strongly during embryogenesis. FLNA and FLNB are ubiquitously expressed throughout the body; however, the expression of FLNC is expressed by skeletal and cardiac muscles [3]. The long-elongated Y-shaped (240–280 kDa) polypeptide chain of FLNA (Figure 1) exists in either homo- or heterodimers and each chain consists of 24 immunoglobulin repeats, which are disrupted by two H1 and H2 domains, whereas H1 comprises between 15 and 16 Ig repeats and H2 between 23 and 24 Ig repeats (Figure 1) [1]. These hinge regions are proteolyzed by calpain and separate 24 Ig repeats into the rod 1 domain, which comprises 1–15 Ig repeats, rod 2 comprising 16–23 Ig repeats and a dimerization domain [4,5]. The hinge regions are proteolyzed by Ca^2+^-dependent calpains, and the cleavage of these sites produces a 90 kDa *C*-terminal fragment of FLNA (FLNA^CT^) and a 200 kDa *N*-terminal fragment (FLNA^NT^) [6].

### 1.1. Filamin A in Cellular Signaling and Migration

FLNA interacts with more than 90 partner proteins to execute multiple cellular functions [7], and mainly helps to provide scaffolding to its interacting partners. Due to the high level of similarity between the Ig repeats, multiple proteins bind at multiple sites of FLNA [8]. Other than providing a structural organization for cells, FLNA protein also mediates organ development, cell signaling, migration, proliferation, cell adhesion, phosphorylation, transcription, and the nuclear transportation of transcriptional factors [5]. 

Cell migration is a crucial step during embryogenesis, wound healing and also remodeling processes, such as myocardial infarction and atherosclerosis. These processes require the specific movement of particular cells to form different tissues. Cell migration is involved in cell polarization, protrusion in the direction of cell movement, retraction, and release from the rear. FLNA is present in most motile cells, at both the leading and rear end, and has been shown to be involved in the remodeling of the actin cytoskeleton, and in cell protrusion and retraction. In addition, FLNA provides scaffolding for multiple cytoskeletal proteins by the integration of cell adhesion [2,4]. FLNA is more localized to lamellipodia, filopodia, stress fibers and focal adhesions [9]. The possible interactions could be due to the specific binding partners of FLNA, such as receptors and adhesion molecules that mostly reside in the cell region under migration [10]. Furthermore, it could also be through higher concentrations of FLNA in newly assembled actin sites in lamellipodia, due to greater avidity for the branched F-actin junctions [11,12]. Furthermore, endoplasmic reticulum stress inducer IRE1α interacts with FLNA to induce the cytoskeletal remodeling and cell migration [13].

### 1.2. Therapeutic Use of Calpain Inhibitors

The H1 and H2 hinge regions of FLNA are proteolyzed by Ca^2+^-dependent calpains that participate in various cellular and physiologic activities, such as cytoskeletal remodeling [14], cell motility, [15], embryonic development [16], signal transduction pathways [17], apoptosis [18], the regulation of gene expression [19], and cell cycles [20]. Chemical calpain inhibitors are generally classified into two groups, i.e., non-peptide calpain inhibitors and peptidomimetic calpain inhibitors [21]. For example, calpain 1 regulates negatively erythrocyte deformability and filtration. As a result, inhibition of calpain 1 has been proposed as a therapeutic approach to treat sickle cell disease [21]. The inhibition of calpain 3 activity treats tibial muscular dystrophy. Either the overexpression or the blocking of calpain 3 activity inhibits disease progression [22]. Both calpain 1 and calpain 2 are extremely abundant in the heart [23]. Despite the enhanced activation of calpains detected during ischemic and reperfusion injury [24], calpain inhibitors in cardiovascular diseases have not been studied in detail. Interestingly, calpain inhibitors have the potential to block the phosphorylation of FAK1, Src, Cdk5 and MAPK kinases that are involved in angiogenesis [25].

## 2. Genetic Disorders Associated with Human Filamin A Mutations

Due to the wide range of function of FLNA in cell migration and cell signaling, mutations in the *FLNA* gene cause a varying spectrum of developmental malformations, which are also associated with cardiovascular malformations and diseases (Figure 2), whereas *FLNB* and *FLNC* mutations are mainly restricted to skeletal and cardiac muscle diseases, respectively. As these proteins have gained interest recently in the research field, their more predominant functions are yet to be explored. The first mutation to be identified in the *FLNA* gene was the null mutation, where coding amino acids are converted to a stop codon in exon 3, resulting in periventricular nodular heterotopia (PVNH), where a six-layer neocortex is not formed due to failed neuronal migration [26]. PVNH is mainly linked to females, while the majority of hemizygous males are confined to embryonic lethality, and live-born males display aortic dilation and die from a massive hemorrhage in the neonatal period [26]. Female PVNH patients run a high risk of strokes and are associated with other cardiovascular abnormalities, such as valvular abnormalities, persistent ductus arteriosus and aneurysms in the aorta [26]. The majority of PVNH patients with *FLNA* mutations also have thoracic aortic aneurysms [27]. Multiple missense mutations, resulting in substitutions in the actin-binding domain (ABD), have been identified, mainly in the Ig repeats of 9, 10, 14, 16, 22 and 23 of FLNA [28]. These mutations are found in otopalatodigital syndrome, Melnick-Needles syndrome and front metaphyseal dysplasia [28], and are also accompanied by cardiac, tracheobronchial and urological malformations, resulting in perinatal death [29]. However, phenotypes caused by the otopalatodigital syndrome, Melnick-Needles syndrome and front metaphyseal dysplasia are very distinct from PVNH [30].

Missense mutations in the *N*-terminal terminal region of the *FLNA* gene are associated with non-syndromic mitral valve dystrophy [31]. The genomic deletion of codons from exon 19 results in cardiac X-linked myxoid valvular dystrophy (XVMD), a specific cardiovascular malformation [32]. XVMD are frequently involved in vascular anomalies, which include mitral valve prolapse, as well as mitral and aortic regurgitation. Interestingly, it is known that a defective signaling cascade in TGF-β results in impaired mitral valve remodeling, and FLNA contributes to changes in cardiac valves by regulating the TGF-β signaling cascade via interaction with SMADs [33]. XVMD are not linked with PVNH and the other congenital disorders mentioned above. Interestingly, a human-induced pluripotent stem cell line from a 10-year-old male patient with cardiac valvular dysplasia, carrying a particular mutation in the *FLNA* gene (c. 84G→A), using non-integrative Sendai virus reprogramming technology, has been established to the study molecular mechanisms of cardiac valvular dysplasia, as this cell line can differentiate into three germ layers in vivo [34].

Exon 21 nonsense mutations, frameshift and 4-shift mutations have been identified in FLNA to cause bilateral PVNH, along with Ehlers-Danlos syndrome, accompanied by minor cardiovascular malformations [35]. A variant of PVNH, associated with *FLNA* mutations and Ehlers-Danlos syndrome, leads to the development of aortic dilation during early childhood [36]. Novel pathogenic variants have been identified in the *FLNA* gene, causing respiratory failure in newborns [37]. In summary, these human *FLNA* mutations are classified as at least moderate pathogenic associations according to the guidelines provided by the American College of Medical Genetics and Genomics [38].

## 3. Mouse Models of Filamin A Deficiency

The discovery of multiple FLNA-deficient melanoma cells has prompted scientists to study the cellular functions in the presence or absence of FLNA [39] and interacting partners of FLNA in further detail. These FLNA-deficient cell lines exhibit continued blebbing of the plasma cell membrane, impaired pseudopod protrusion, and cell migration. When FLNA deficiency has been restored in these cell lines, by transfecting the full-length coding sequence of FLNA, cell migration has been restored, thereby leading to faster migration than FLNA-deficient cells in response to a chemoattractant [39]. 

The mouse has become the default animal for studies of the biology of FLNA because there is a vast scientific database and a large set of reagents, and because it is the least expensive animal model as compared to other small rodents such as rats, to larger species such as guinea pigs, chickens, ferrets, and nonhuman primates. As the ability to modify the mouse genome has made genetically engineered knockout mice a powerful tool for modeling human disease and there are a wide variety of reagents and tools available for mice, a few mouse models of filamin deficiency have been used. First, two different mouse models deficient for FLNA have been introduced, one of which is chemically induced [40] and another that is a genetically modified model [41]. A nonsense mutation has been introduced in Ig repeat 22 (Y2388X), designated as *Dilp2* in the chemically induced model, and leading to the loss of FLNA function [40]. This mutation, induced by *N*-ethyl-*N*-nitrosourea, results in the conversion of tyrosine to stop codon, causing lethality, due to the arterial trunk, midline fusion defects, and palate abnormalities in male mice [40]. To study the tissue-specific function of FLNA, a genetically modified mouse has been generated by introducing flox sites in the *FLNA* gene, between introns 2 and 7 in female mice. The crossbreeding of these female mice with β-actin Cre male mice results in earlier truncation at the 121 amino acid in the CH1 domain, causing embryonic lethality in males, with severe cardiovascular abnormalities and abnormal vascular patterning [41].

In the following years, more FLNA-deficient mouse lines have been developed to study the distinct roles for FLNA in a specific cell manner. Mice that are deficient in FLNA in vascular smooth muscle cells (VSMCs) have lower blood pressure, due to a decreased pulse rate, aortic dilation and an increase in atrial compliance [42]. Megakaryocyte-specific deletion of FLNA in mice results in severe macrothrombocytopenia, due to accelerated platelet clearance, accompanied by lethality in late embryogenesis [43]. The FLNA-deficient megakaryocyte cells are prematurely large and have fragile platelets, which are quickly cleared by macrophages in circulation [43]. Mice that are deficient in FLNA in endothelial cells, by using Cre expression driven by the vascular endothelial-cadherin promoter show no abnormalities in vasculature or heart pump function, but, when myocardial infarction (MI) is induced in these mice, the endothelial cells in FLNA-deficient mice fail to generate new blood vessels, which results in larger scar areas after MI [44]. Nor do mice that are deficient in FLNA in monocytes show any abnormalities by using Cre expression driven by the lysozyme-M promoter [45]. They are fertile and viable. However, FLNA-deficient monocytes fail to migrate poorly, which is partly controlled by Rho GTPase activation. This indicates that FLNA is required at the earlier stages of macrophage differentiation [46]. We deleted FLNA in macrophages and observed no phenotype in the mice, but when atherosclerosis is induced as a result of a western diet, mice deficient in FLNA in macrophages develop smaller plaques, due to impaired macrophage cell migration, lipid uptake and cytokine response [45]. 

Earlier, we reported the hypoxia-inducible factor-1α (HIF-1α) as a novel interacting partner of FLNA [47]. During hypoxia, hydroxylation is inhibited, resulting in the stabilization of HIF-1α, which is then translocated into the nucleus to activate its target genes, including vascular endothelial growth factor-A (VEGF-A) [48]. We have reported that a cleaved *C*-terminal fragment of FLNA (FLNA^CT^) promotes the transcriptional activity of HIF-1α by facilitating nuclear translocation, and induces the secretion of VEGF-A [47]. Following the induction of MI in mice that are deficient in FLNA in endothelial cells (ECs), we observed lower serum levels of secreted VEGF-A, larger scar size and enlarged hearts [44]. These findings indicate that FLNA deficiency in ECs fails to generate new blood vessels in the peri-infarcted areas of the cardiac muscle.

## 4. Filamin A in the Heart

FLNC is highly expressed in cardiac muscles and its mutations are associated with fewer phenotypes, such as human hypertrophic and restrictive cardiomyopathy [49], with a higher incidence of sudden cardiac death [50]. In contrast to FLNC, FLNA is expressed in virtually every human tissue, and mutations in the *FLNA* gene result mainly in congenital malformations, along with cardiac phenotypes such as myxomatous mitral valve disease [48], non-syndromic mitral valve dystrophy [31], familial Ebstein’s anomaly, a rare form of congenital heart disease [51], prolapsed atrioventricular valves [52], and also in FG syndrome, a rare syndromic cause of X-linked mental retardation associated with congenital heart disease [53]. Thus, screening for *FLNA* mutations is recommended in cardiac anomalies, including familial myxomatous valvular dystrophy, particularly if X-linked inheritance is suspected. Experimentally, FLNA-null embryos display abnormal epithelial and endothelial organization, and aberrant adherens junctions in developing blood vessels and the heart [41]. 

The essential roles for FLNA in intercellular junctions provide mechanisms for the diverse developmental defects seen in patients with *FLNA* mutations [41]. In mitral valve disease, the loss of FLNA function results in increased extracellular matrix production [54]. The interaction of FLNA with Asb2α is essential for actin cytoskeleton remodeling during heart development [55]. Mutations in *FLNA* have profound effects on membrane excitability, as Ca^2+^-activated K^+^ channels have been shown to play critical roles in shaping the cardiac atrial action potential profile. FLNA augments the trafficking of these channels in cardiac myocytes [56]. Thus, changes in this channel trafficking would significantly alter atrial action potential, and consequently atrial excitability. Furthermore, mutations in *FLNA* are frequently associated with severe arterial abnormalities. It seems that FLNA is a critical upstream element of the signaling cascade underlying the myogenic tone [57]. The deletion of smooth muscle-specific FLNA in mice recapitulates the vascular phenotype of human bilateral PVNH, culminating in aortic dilatation [42]. This suggests a brain and cardiovascular connection through FLNA in multiple anomalies, including PVNH.

## 5. Filamin A in the Arterial Wall 

### 5.1. Filamin A in Vascular Smooth Muscle Cells

Filamin was first discovered and isolated in chicken gizzard VSMCs and non-smooth muscle cells in 1975 [58]. A large body of the following literature has addressed the morphology and function of filamins in this tissue. However, the distribution of filamins in aortic VSMCs differ from those in gizzard VSMCs, and even femoral artery VSMCs, underscoring the importance of studying filamins in a context-specific manner [59]. The roles of filamins in actin cross-linking and membrane stabilization do not differ in VSMCs and other tissue SMCs. However, the tissue-specific distribution of filamins may be partly owing to their interaction with diverse transmembrane and signaling proteins, and thus affects the functions of filamins in different tissue SMCs.

VSMCs form the medial layer of arteries such as the aorta, which confers elasticity and strength to the blood vessel wall (Figure 3). FLNA physically links the integrin receptors and VSMC contractile filaments and contributes to the integrity of the aortic wall [60]. The VSMC layer contracts or relaxes to regulate blood volume and pressure, which is initiated by mechanical, electrical, and chemical stimuli. The contraction of VSMC is regulated by the sympathetic nervous system through adrenoceptors. A specific subtype of adrenoreceptors, α_2_C-adrenoceptors (α_2_C-ARs), mediates the vasoconstriction of small blood vessels, particularly arterioles. With a series of studies and in silico modeling, FLNB interacts with α2C-ARs in arteriolar SMCs. This interaction is essential to the translocation of α_2_C-ARs from the perinuclear region to the cell surface where they react to stress. The mechanism involves the phosphorylation of FLNB at Ser2113 by cAMP–Rap1A–Rho–ROCK signaling [61,62,63]. In addition to their response to nerve innervation, VSMCs can also initiate contraction by themselves, in a process termed myogenic response, where the VSMC membrane depolarizes in reaction to the mechanical stretching of the muscle. The electrical depolarization of VSMCs is linked to the opening of the mechanosensitive and nonselective stretch-activated cation channels (SAC), followed by the opening of voltage-gated calcium channels, resulting in an increase in intracellular Ca^2+^ concentration and myocyte constriction. One of the regulatory mechanisms of this process is the change in polycystin ratio, TRPP1/TRPP2, which regulates the activity of native SACs. FLNA interacts with TRPP2 and mediates the inhibitory function of TRPP2 on SAC activity [64]. Furthermore, the inwardly rectifying potassium channels are also sensitive to flow, and the isoform Kir2.1 is expressed in VSMCs, acting as an important regulator of vascular tone. The surface expression and location of Kir2.1 is regulated by the direct interaction with FLNA in arterial VSMCs [65]. Therefore, filamins not only maintain the resting blood vessel wall integrity, but also mediate the vasoconstriction signaling that regulates vascular resistance upon stimulus.

In pathological conditions, such as atherosclerosis and restenosis, VSMCs switch from the contractile phenotype to the proliferative, synthetic, migratory, and/or macrophage-like phenotypes. The G-protein-coupled P2Y2 nucleotide receptor (P2Y2R) is an activator of the phenotypic switching. The binding of FLNA to P2Y2R has an essential role in the P2Y2R agonist UTP-stimulated spreading and migration of VSMCs [66] (Figure 3), the secretion of the proinflammatory cytokine lymphotoxin-α [67], and the low-density lipoprotein receptor-related protein-mediated internalization of aggregated low-density lipoproteins [68]. Tissue factor signaling was also found to promote VSMC migration in addition to its pro-coagulant activity. VSMCs in the hyperplasic intima express tissue factor in human coronary artery atherosclerotic plaques obtained from the hearts of ischemic heart disease [69]. 

In cultured VSMCs, FLNA colocalizes with tissue factor at the migratory front and mediates signaling [69]. In accordance, when FLNA was silenced in VSMCs by shRNA, the pathologic phenotype switch of VSMCs exposed to ox-LDL was attenuated and the injury to the cytoskeleton was less prominent [70]. Given the impact on VSMC phenotypic switching, one may speculate that FLNA may be upregulated in the plaque and contribute to the atherosclerotic progression. Surprisingly, in a proteomic study searching for VSMC-specific protein alterations, FLNA has been found to be downregulated in the media layers isolated from human atherosclerotic coronary artery by laser microdissection, a finding further validated by immunohistochemistry [71]. VSMCs undergo multiple processes, including phenotypic switching as well as apoptosis in atherosclerosis, which may occur at different stages or in different regions of the plaque. The additive effect of these processes can be either promoting plaque formation or stabilizing plaques and preventing the rupture of the fibrous cap [72]. The VSMC-specific knockout mouse model may be necessary to elucidate the overall effect of VSMC-specific FLNA in atherosclerosis.

### 5.2. Filamin A in Endothelial Cells

ECs play vital roles in the pathogenesis of many stages of atherosclerosis [73]. It has been reported that FLNA is expressed by vascular EC to mediate cell motility [74] (Figure 3). A lack of FLNA in endothelial cells impairs tubular formation and migration [41]. FLNA and migfilin bind to integrins to activate the integrins, and this activation is required to regulate the cell adhesion and migration in endothelial cells [75]. FLNA expression is required for the α subunit of the epithelial sodium channel and stress fiber formation to maintain the capillary barrier function [76]. FLNA interacts with R-RAS to maintain the endothelial barrier function, and a loss of FLNA resulted in increased vascular permeability [77]. Phosphorylation of FLNA is required to target the membrane adenylyl cyclase activity in the endothelial barrier [78]. Furthermore, FLNA interacts with endothelial-specific molecule-2 to mediate the cell chemotaxis and tube formation [73]. Cavolae mediates the endothelial endocytosis and trafficking by recruitment of dynamin-2 and FLNA [79]. FLNA binds to the cytoplasmic domain of tissue factor to facilitate incorporation of tissue factor into cell-derived microvesicles [80]. In vertebrate multiciliated cells, FLNA interacts with R-ras near the basal bodies to promote multiciliogenesis [81] (Figure 3). 

FLNA-null embryos indicate irregular endothelial organization and abnormal adherent junctions in developing organs, including blood vessels and the heart [41]. Mice that are deficient for FLNA in ECs are born normally without cardiovascular malformation [74]. However, the endothelial response, when stressed for example during tumor progression, is greatly impaired, leading to smaller tumor growth in mice deficient for FLNA in ECs [74]. Interestingly when these mice were induced with MI, they developed left ventricular (LV) dysfunction and cardiac failure, due to defective endothelial response impaired signaling in VEGF-A, AKT, ERK1/2 and GTP RAC1 [44]. The cleaved 90 kDa FLNA^CT^ fragment is essential for mediating cellular signaling and the transportation of transcription factors into the cell nucleus [47]. The inhibition of FLNA^CT^ production by calpeptin in ECs reduces proliferation and migration [82]. Similar to FLNA, FLNB is also expressed by ECs and its deficiency leads to impaired development of microvasculature [83]. 

A lack of FLNA in ECs increases the infarction size after MI, probably due to impaired EC function and signaling [44]. This can be explained by three major factors. First, the lack of FLNA in ECs results in impaired remodeling of the LV, accompanied by an increased infarction size and a thinned and enlarged LV wall. Second, a fewer number of capillaries were observed in the peri-infarcted areas. Third, ECs deficient in FLNA display impaired cellular signaling, resulting in reduced cell migration and tube formation. Cardiac parameters are not altered in response to dobutamine stress in mice deficient in FLNA in ECs. Furthermore, the serum levels of NT-proBNP, a marker of heart failure, are higher in these mice, indicating an increased ventricular response to volume expansion and possibly increased wall stress [44]. 

The insides of the heart chambers are covered by the endocardium, which consists of ECs. The regenerating capability of cardiomyocytes is very limited, and advanced atherosclerotic plaques reduce blood circulation in areas with cardiomyocytes requiring nutrients. To support cardiomyocytes during MI, ECs are essential in angiogenesis, by active migration and proliferation. Interestingly, we only observed a reduced number of capillary structures in infarcted areas. However, the number of capillary structures in non-infarcted areas remains unchanged. This suggests that angiogenesis regulated by FLNA was induced when these mice were exposed to a stressed condition. Without stress, these mice did not exhibit any phenotype. We have shown no difference in EC proliferation, in contrast to fibroblasts, in which the absence of FLNA reduces cellular proliferation. However, the impaired migration of FLNA-deficient ECs observed in vitro may explain the increased size of MI in these mice. VEGF-A is a cardinal player in regulating angiogenesis during MI [84]. VEGF-A binds to VEGF receptor 2 to regulate downstream signaling molecules, such as AKT, ERK and RAC-1 GTPase, during angiogenesis [85]. From our previous findings, we report that FLNA^CT^ interacts with HIF-1α and regulates VEGF-A under hypoxia in multiple human cancer cells [47]. Interestingly, the serum levels of VEGF-A were decreased in mice deficient in FLNA in endothelial cells after MI. The phosphorylation of kinases, such as AKT and ERK1/2, and the level of active GTPase RAC1 are also reduced in cultured endothelial cells in the absence of FLNA (Figure 3). It is known that FLNA induces cellular migration, either by direct interaction with RAC1 [46] or by the indirect regulation of VEGF-A [47]. In our studies, the absence of FLNA in endothelial cells increases the size of the LV, decreases cardiac pump function, and leads to the development of cardiac failure by blunting the angiogenic responses that are required for satisfactory wound healing [44]. Inducing cellular signaling pathways regulated by FLNA and related to angiogenesis during MI could be a new approach to reduce the size of MI.

### 5.3. Filamin A in Macrophages

Macrophages play a vital role in the disease progression of atherosclerosis partly by engulfing the lipids. This leads to a heart attack and stroke, due to the disrupture of the plaque or the narrowing of the blood vessels. FLNA was first discovered in the macrophages of chicken gizzard, and among all the three isoforms, FLNA is predominantly expressed in macrophages [58]. The deletion of macrophage-specific FLNA or the inhibition of FLNA^CT^ by calpeptin in mice reduces the atherosclerosis [45] explained by three main observations. Firstly, FLNA-deficient macrophages migrated and proliferated poorly, secondly FLNA-deficient macrophages secreted lower levels of inflammatory cytokine IL6, which is partly regulated by STAT3 transcription factor, and FLNA-deficient macrophages uptake lipids poorly [45]. FLNA-deficient monocytes fail to migrate, which might be controlled by Rho GTPase activation, indicating that FLNA might be required in the earlier stages of macrophage differentiation [46]. The binding of P2YR2R to FLNA is important for P2YR2R activation and inducing the agLDL uptake in VSMCs [68]. Inflammasome activation induces the activation of intracellular caspase 1/calpain, which degrades FLNA and serves as bonds between the cytoskeleton and tissue factor, and promotes the coagulation activation shown in macrophages [86]. 

The FLNA-deficient megakaryocyte cells are prematurely large and have fragile platelets, which are quickly cleared by macrophages in circulation [43]. FLNA interacts with FcgR1 at the plasma membrane in monocytes, and this interaction promotes FcgR1 sub-surface expression by rerouting or retention from the lysosomal pathway [87]. FLNA acts as a checkpoint in cartilage repair, by interacting with CBFβ and inhibiting chondrogenesis. When the chemical substance kartogenin is introduced, it binds to FLNA and disrupts CBFβ interaction, and it promotes chondrogenesis by regulating the CBFβ–RUNX1 transcriptional program [88]. Mutations in both *FLNA* and *FLNB* are associated with chondrodysplasias, as a compensatory mechanism of either isoform is required to regulate the integrin expression as well as RhoA activation during actin stress fiber formation and remodeling [89]. FLNA interacts with CCR2B at the cell surface and in the internalized vesicles; this interaction helps in localizing the CCR2B protein to spatial localization in various dynamic membrane structures [90]. FLNB was shown to interact with uPAR to regulate the Serp-1 inhibition in monocyte adhesion, and the loss of FLNB blocked the Serp-1 to monocyte adhesion [91]. Interestingly, FLNA interacts with RAC1 in macrophages and protects against atherosclerosis [92].

A lack of FLNA in macrophages results in a reduced size of atherosclerotic plaques, due to impaired macrophage function and signaling (Figure 3). During the initiation and progression of atherosclerosis, macrophages proliferate, migrate, form foam cells, and secrete inflammatory substances [93]. Following bone marrow transplantation of FLNA-deficient monocytes/macrophages, we have shown the reduced size of atherosclerotic plaques in atherogenic *Ldlr*^−/^^−^ mice [45]. Furthermore, mice that are deficient in FLNA in macrophages display a reduced size of atherosclerotic plaques after overexpressing PCSK9 by an adenoviral vector. We hypothesized that this plaque reduction could be based on three major findings. First, FLNA-deficient macrophages produce lower levels of the pro-inflammatory cytokines IL-6 and IL-12, which could be partly regulated by STAT3. Second, FLNA-deficient macrophages proliferate and migrate less. Third, FLNA-deficient macrophages form foam cells less and excrete more cholesterol. Together, these could explain the reduced size of the atherosclerotic plaque in atherogenic mice who are deficient in FLNA in macrophages. In vitro, macrophages without FLNA proliferate and migrate less. This could be partly due to lower p-AKT and p-ERK1/2 in the regulation of macrophage proliferation and migration, similar to endothelial cells [45]. 

Macrophages produce inflammatory cytokines in response to external stimuli during atherosclerosis. Anti-inflammatory therapies reduce atherosclerosis, and the treatment of atherogenic *Ldlr*^−/^^−^ mice with antibodies against the IL-6 receptor reduces the size of the atherosclerotic plaques [94]. We observed a reduction in the secretion of IL-6 and IL-12 in FLNA-deficient macrophages in vitro, and in the blood serum from mice that are deficient for FLNA in macrophages. STAT3 partly regulates IL-6 secretion by binding to its promoter region [95]. In our in vitro studies, FLNA^CT^ produced by calpain cleavage binds to STAT3 and this complex is translocated into the nucleus to regulate the expression of IL-6. Furthermore, a deficiency of FLNA in macrophages inhibits the nuclear p-STAT3 in vitro. Interestingly, blocking FLNA^CT^ production by calpeptin reduces the levels of nuclear p-STAT3, followed by the secretion of IL-6 in macrophage, which controls to levels similar to those detected in FLNA-deficient macrophages. This suggests that the cleavage of FLNA by calpain is important for STAT3 signaling, by increasing the nuclear levels of p-STAT3 and thereby increasing the inflammatory response by IL-6 in macrophages. 

Lipid uptake and the excretion of lipids from macrophages are the main vascular events in the formation of atherosclerotic plaques [45]. We observed that FLNA-deficient macrophages take up fewer lipids and excrete more cholesterol. To explain these functional changes, we studied the signaling molecules that are involved in the unloading of cholesterol from macrophages without FLNA, and discovered higher expression protein levels of CD36, SR-B1, and COX2. Interestingly, control macrophages treated with calpeptin reduced foam cell formation as well as lipid uptake. Finally, the systemic administration of calpeptin reduces the size of atherosclerotic plaques in atherosclerotic control mice, to levels similar to those we observed in mice deficient in FLNA in macrophages in vivo. In summary, either a deficiency of FLNA in macrophages or blocking production of FLNA^CT^ by calpeptin reduces the proliferation, migration, and secretion of inflammatory cytokines, and the formation of foam cells in vitro. These inhibitory effects may partly explain the reduced size of the atherosclerotic plaques in mice that are deficient in FLNA in macrophages. Thus, we propose a causal link for FLNA-dependent macrophage cell function during atherogenesis.

### 5.4. Filamin A in T Cells

Atherosclerosis is a slow-progressing chronic inflammatory disease. Recent studies from in vivo imaging, mRNA sequencing, and knockout mouse models indicate T cells as crucial drivers in atherosclerosis [96]. Among the three isoforms of filamins, FLNA is predominantly expressed in T cells, and several studies indicate that FLNA plays a crucial role in T-cell activation (Figure 3). Integrins are required for T-cell trafficking from the bloodstream into various tissues, and FLNA is required for the formation of strong integrin ligand bonds. The absence of FLNA results in reduced integrin ligand bonds, adhesion and trafficking into sites of inflammation under sheer stress [97]. The phosphorylation of the β2 integrin at the Thr758 site inhibits FLNA binding and allows 14-3-3 to bind to the integrin to promote cell adhesion [98]. Serine/threonine kinase NDR2 phosphorylates FLNA at serine 2152, this phosphorylation results in the dissociation of FLNA with integrin LFA-1, and the association with talin and kindlin-3 to stabilize the open conformation and activation of LFA-1 [99]. The binding of Rap1–GTP to FLNA weakens the β2chain in LFA-1, which also results in LFA-1 activation [100]. CD28 is a key regulator for inducing cytoskeletal changes and lipid raft accumulation. CD28 interacts with FLNA and recruits FLNA to immunological synapses, where it organizes CD28 signaling [101]. DeSUMOylation inhibited the binding of PKCθ with CD28 and FLNA, which resulted in impaired mature immunological synapses [102]. The antigen stimulation of T cells recruits FLNA to the T-cell APC contact area, where it co-localizes PKCθ to induce T cell activation and IL-2 production [103]. FLNA interacts with NF-kB-activating kinase (NIK) to form a complex in the absence of the T-cell receptor. The FLNA–NIK complex induces IKKα activity [104]. C-MIP and TC-mip, a splice variant of C-MIP, are key regulators in Th2 signaling in patients with minimal-change nephrotic syndrome (MCNS). FLNA interacts with both C-MIP and its splice variant, and, moreover, FLNA expression is elevated in MCNS patients, indicating FLNA/C-MIP signaling could be a potential target for treating MCNS [105].

## 6. Future Directions

In addition to its well-known cytoskeletal function, FLNA performs multiple functions in cellular signaling and transcription. In this review, we updated some novel functions of FLNA, particularly related to cardiovascular remodeling after MI and atherosclerosis (Figure 4). FLNA plays a vital role during embryogenesis, as a deficiency in FLNA is associated with lethality, due to various devastating malformations affecting the development of the heart, skeleton and brain in humans. Similarly, mice that are deficient in FLNA die in utero due to severe cardiovascular malformations. Experimental studies on the cell-specific expression of FLNA provide new insights into the cardiovascular diseases. The expression of FLNA in VSMCs acts as an important regulator of vascular tone. Endothelial FLNA decreases the size of MI and protects against cardiac failure. 

During remodeling after MI, angiogenesis is the crucial repair mechanism, in which FLNA induces angiogenesis by upregulating the VEGF pathway to gradually protect against heart failure. FLNA may be locally overexpressed in coronary endothelial cells to promote angiogenesis, using cationic liposomes during percutaneous transluminal coronary angioplasty, to repair the infarcted myocardium. 

Our discovery on the novel interaction of FLNA^CT^ with STAT3, in regulating the inflammatory signaling cascade as an example, identifies FLNA as an interesting target. In this regard, the design and screening of small molecules to inhibit the cleavage or expression of FLNA^CT^ and/or use of protein–protein inhibitors to specifically block the interaction of FLNA with its partners, will reduce inflammation and lipid storage in macrophages. This design could serve as a treatment modality against atherosclerosis. Screening the presence and nature of *FLNA* mutations in humans and identifying mutations that abolish the binding ability of interacting FLNA partners at different positions would be interesting to understand the biology behind the serious cardiovascular malformations. If such critical *FLNA* mutations are identified, patients can be informed that they run a higher risk of developing cardiovascular malformations, and can therefore be closely followed up by routine biochemical and radiological examinations. Calpain inhibitors are widely used in various diseases, but they have not been studied in detail in cardiovascular cells and disease remodeling. Blocking the production of FLNA^CT^ by calpeptin treatment impairs macrophage cell proliferation, migration, and lipid uptake, as well as atherosclerotic plaque size in vivo. These results indicate that the inhibition of calpain might serve as a potential target to treat or at least slow atherogenesis.

## Figures and Tables

**Figure 1 ijms-22-06555-f001:**
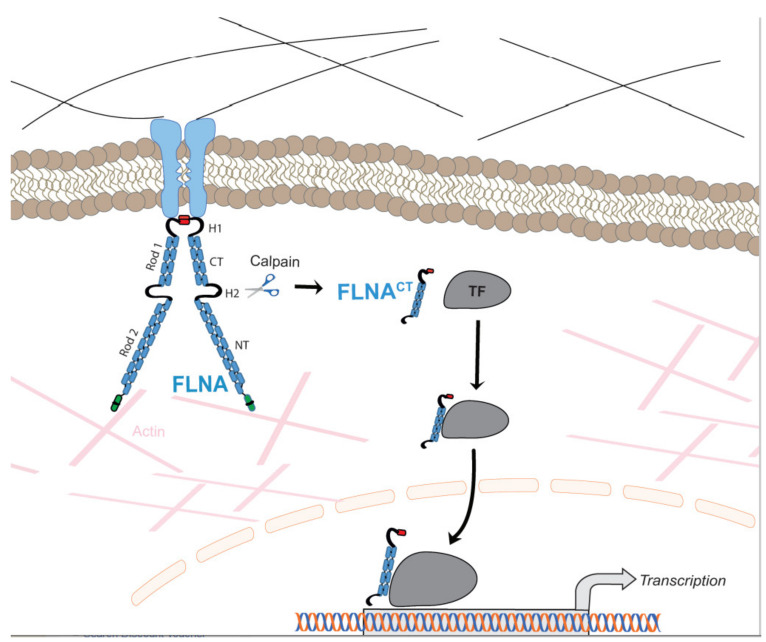
Schematic illustration of the regulation of function of transcriptional factors (TF) by FLNA. Membrane-bound FLNA mediates extracellular signals to the cytoskeleton. FLNA interacts with multiple TF in a cell-specific manner in the cytoplasm. Increased calpain protease activity cleaves membrane-bound FLNA releasing the FLNA^CT^, which is translocated to the nucleus together with TF. As part of the transcriptional complex, FLNA^CT^ can also bind to promoter regions of target genes. Thus, FLNA^CT^ increases the transactivation function by facilitating translocation to the nucleus and/or nuclear retention and/or by working as a transcriptional coactivator.

**Figure 2 ijms-22-06555-f002:**
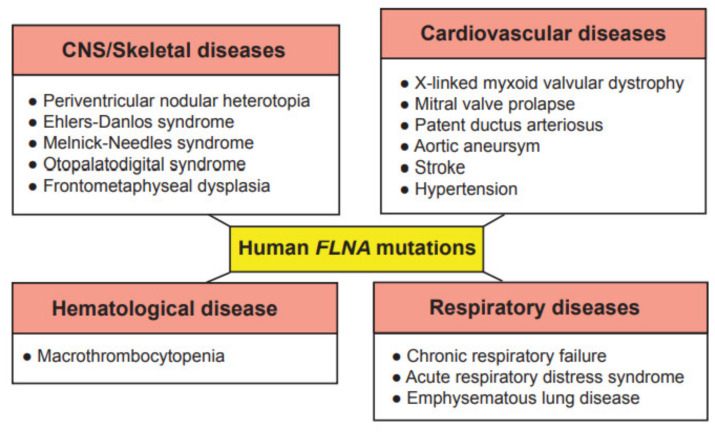
Genetic diseases associated with mutations in the human *FLNA* gene. These genetic diseases are grouped into central nervous system (CNS)/skeletal, cardiovascular, hematological and respiratory diseases.

**Figure 3 ijms-22-06555-f003:**
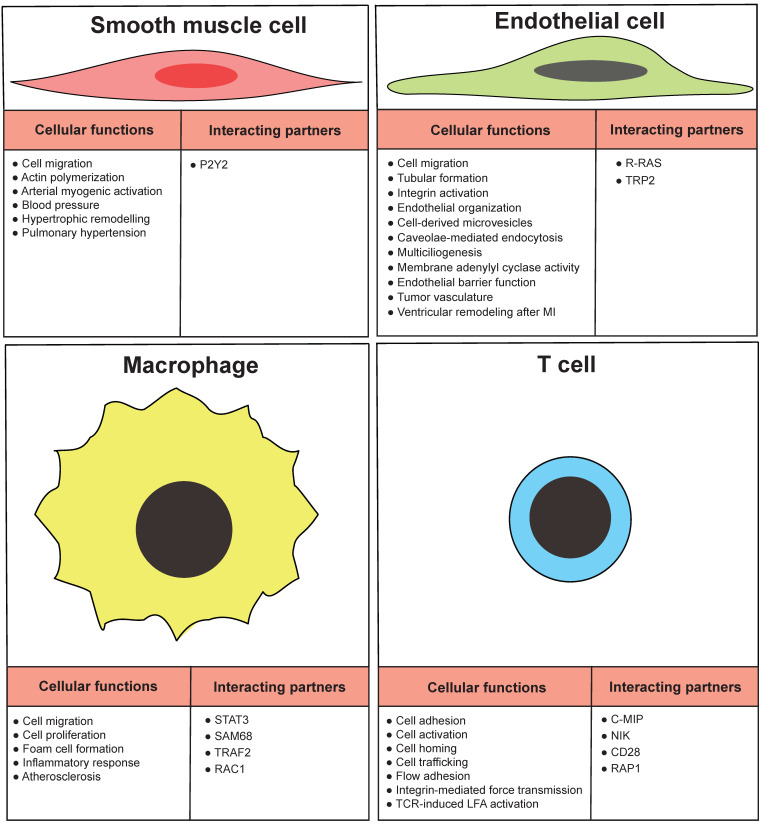
Functions of arterial wall cells regulated by FLNA. Cellular functions and some diseases are presented in relation to vascular smooth muscle cells, endothelial cells, macrophages and T cells. Furthermore, some interacting partners of FLNA in particular cell type are listed.

**Figure 4 ijms-22-06555-f004:**
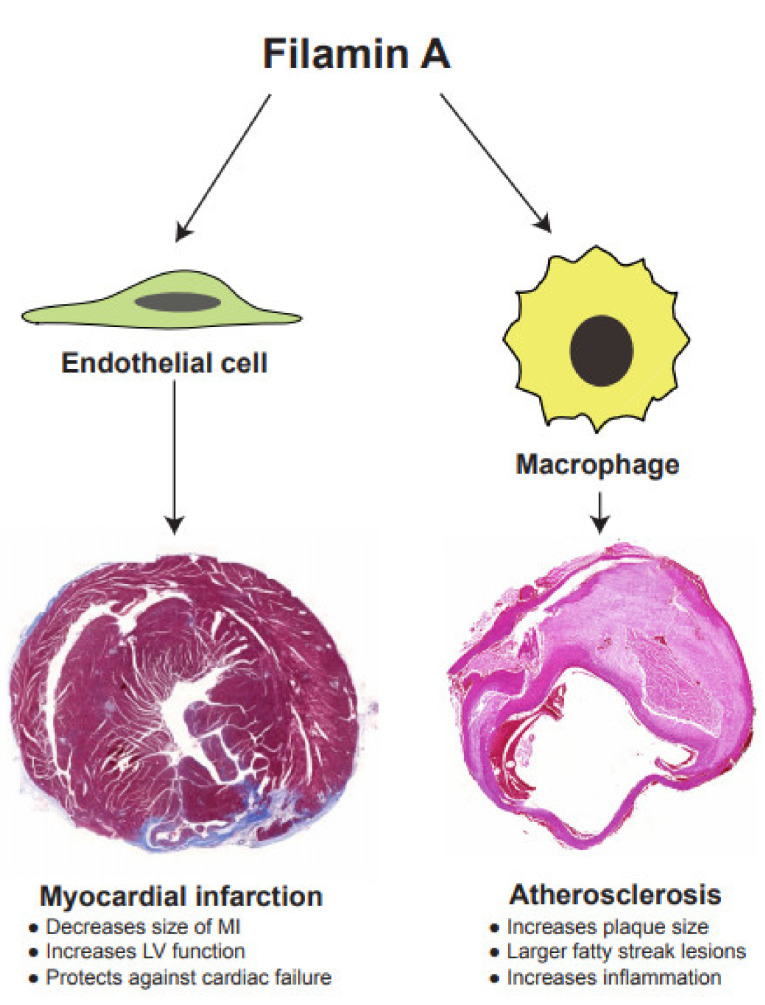
FLNA regulates functions of both endothelial cells and macrophages that are critical for the progression of myocardial infarction and atherosclerosis, respectively.

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
