# Peer review of "Filamin A Regulates Cardiovascular Remodeling"

_ijms, 2021, doi:10.3390/ijms22126555_

Round 1

Reviewer 1 Report

The review by Bandaru et al provides an examination on the role of Filamin A in cardiovascular remodeling, reviewing recent literature on their role in the heart and in the arterial wall. Although the data described and discussed are very informative, some explanations are confusing. In order to transform this work into a more attractive one, I suggest to carefully check english structure and style.

1. Some sentences lack any verb or use incorrect ones. Here you can find some examples, although others can be found through the manuscript:

  • Lines 84-85: “…calpain 1 is negatively regulates of erythrocyte deformability”
  • Line 149: “…due to arterial ttrunk,…” (the verb seems to lack).
  • Page 170: “…fail to develop poor monocyte migration…” seems an strange construction.
  • Lines 184-186: sentence difficult to understand.
  • Line 233: “FLNB interacts with” should be change by “was shown to interact with”
  • Line 284: This sentence seems to be poorly constructed.

2. In addition some parts of the text are repeated twice. Lines 309 to 321 are repeated in page 5. The same happens with lines 333-335.

3. Some of the references are not correctly mentioned. In lines 86-87, authors state that “calpain 3 inhibition is beneficial in treating tibial muscular distrophy. Either the overexpression or the inhibition of calpain 3 inhits disease progression [21]”. However, in that reference the effects of calpain inhibition were not linked to FLNA. Similarly, whereas authors suggest that higher concentrations of calpains are detected during ischemi and reperfusion injury [23], in that paper it was demonstrated an enhanced activation, but no changes in total calpain levels.

4. Line 167: I suggest to modify that sentence. Instead of “…which results in larger infarction size [45]”, it would be better “…which results in larger scar areas after MI”. 

Author Response

Reviewer #1

Lines 84-85: “…calpain 1 is negatively regulates of erythrocyte deformability”

This sentence has been re-written as suggested (lines 82-83, page 3).

Line 149: “…due to arterial trunk,…” (the verb seems to lack).

The verb has been added as suggested (lines 160-162, page 4).

Page 170: “…fail to develop poor monocyte migration…” seems an strange construction.

This sentence has been re-constructed as suggested (lines 181-183, page 5).

Lines 184-186: sentence difficult to understand.

This sentence has been re-written as suggested (lines 194-197, page 5).

Line 233: “FLNB interacts with” should be change by “was shown to interact with”

These words have been replaced as suggested (lines 384, page 9).

Line 284: This sentence seems to be poorly constructed.

This sentence has been re-constructed as suggested (lines 410-411, page 10).

In addition some parts of the text are repeated twice. Lines 309 to 321 are repeated in page 5. The same happens with lines 333-335.

These repeated sentences have been deleted as suggested.

Some of the references are not correctly mentioned. In lines 86-87, authors state that “calpain 3 inhibition is beneficial in treating tibial muscular distrophy. Either the overexpression or the inhibition of calpain 3 inhits disease progression [21]”. However, in that reference the effects of calpain inhibition were not linked to FLNA. Similarly, whereas authors suggest that higher concentrations of calpains are detected during ischemi and reperfusion injury [23], in that paper it was demonstrated an enhanced activation, but no changes in total calpain levels.

Information on these references has been better clarified as suggested (lines 82-83 as well as lines 85-89, page 3). In this part, we provided calpain functions without relation to FLNA.

Line 167: I suggest to modify that sentence. Instead of “…which results in larger infarction size [45]”, it would be better “…which results in larger scar areas after MI”. 

These words have been replaced as suggested (lines 179, page 5).

Reviewer 2 Report

Authors performed an exhaustive revison of FLNA.

Only few points should be completed:

  • Animal model data is focused on mice. Any other?
  • Any data concerning hiPSC?
  • Please clarify if diseases reported in the manuscript are definitely associated with FLNA mutations or potentially associated.
  • Please add data concerning FLNA mutations definitevely classified as Pathogenic/Likely Pathogenic following ACMG guidelines in contrast to rare variants potentially deleterious but currently classified as VUS.

Author Response

Reviewer #2

Animal model data is focused on mice. Any other?

This information has been provided as suggested (lines 150-156, page 4).

Any data concerning hiPSC?

This information has been re-written as suggested (lines 123-128, page 3). A new reference on this report has been added as reference# 105.

Please clarify if diseases reported in the manuscript are definitely associated with FLNA mutations or potentially associated.

This statement has been clearly written as suggested (lines 134-136, page 4).

Please add data concerning FLNA mutations definitevely classified as Pathogenic/Likely Pathogenic following ACMG guidelines in contrast to rare variants potentially deleterious but currently classified as VUS.

This has been corrected as suggested (lines 134-136, page 4) and a new reference# 106 has been added.

Reviewer 3 Report

Filamin A is an actin binding protein and plays an important role in cell migration. It was reported that Filamin A mutations caused various cardiovascular and lung diseases, including valvular abnormalities. In this review, Bandaru, et al. summarized Filamin A’s cellular functions in cell migration and Filamin A mutations in human disease. Importantly, the authors focused on Filamin A’s functions in the different types of cells in the cardiovascular system, including vascular smooth muscle cells, endothelial cells and blood cells. In general, the manuscript is well organized and clearly presented. The reviewer only has two minor concerns as followed

  • The recent five years references are less than 15%. It is better to add some new references into the manuscript, such as NCB, 2018.20:942
  • Akyurek is a seasoned scientist in the study field of Filamin A. It is better to add a section to discuss the import unaddressed questions in Filamin A study field to attract more newcomers.

Author Response

The recent five years references are less than 15%. It is better to add some new references into the manuscript, such as NCB, 2018.20:942

Multiple recent references have been provided such as ref# 7 (line 57, page 2), ref# 35 (line 137, page 4), ref# 40 (line 176, page 5) and ref# 50 (line 208, page 5).

As suggested, new information (NCB, 2018.20:942) has been added as ref# 107 (lines 76-77, page 2).

Akyurek is a seasoned scientist in the study field of Filamin A. It is better to add a section to discuss the import unaddressed questions in Filamin A study field to attract more newcomers.

We replaced the last subtitle “Concluding remarks” with “Future directions” (line 464, page 11).

As suggested, additional unaddressed points in this field have been proposed (lines 478-482 and lines 484-488, pages 11 and 12).

Round 2

Reviewer 1 Report

Although the review by Bandaru et al can be of interest, it is poorly structured, a little bit confusing, and not centered in the aim of the manuscript. I would suggest to rewrite it with more care.

Author Response

Please see our earlier response to the reviewers.